

# Investigating the relationship between intergroup physical contact and attitudes towards foreigners: the mediating role of quality of intergroup contact

Soraya E. Shamloo[*], Andrea Carnaghi[*] and Carlo Fantoni

Department of Life Sciences, University of Trieste, Trieste, Italy
[*] These authors contributed equally to this work.

## ABSTRACT

Recent research has shown that a brief, casual touch administered by an outgroup member reduces prejudice towards the group to which the toucher belongs. In this study, we take the research on physical contact and prejudice a step further by addressing the relation between individuals' amount of Experienced Intergroup Physical Contact (EIPC), across distinct contexts and involving different body parts, and attitudes towards foreign people. Specifically, we hypothesized that the amount of EIPC would be positively associated with both quantity and quality of intergroup contact, but that only quality would mediate the relationship between the amount of EIPC and outgroup attitudes, quality being more directly linked to the evaluative component of outgroup attitudes. To attain this aim, we asked participants to self-report the amount of EIPC, the quantity and quality of their intergroup contact and their attitudes towards foreign people. Consistent with our hypothesis: (1) as EIPC increased, positive attitudes towards foreign people increased; (2) higher levels of EIPC were associated with better quality and higher quantity of intergroup contact; (3) only quality of intergroup contact mediated the relationship between the amount of EIPC and attitudes towards foreign people. Results were discussed in relation to research on intergroup contact and physical contact.

## INTRODUCTION

Challenges raised by conflicting intergroup relations have always been of primary interest for social sciences. Given the growing anti-immigration public opinion (e.g., *Blinder & Allen, 2016*), research focused on finding ways to promote harmonious intergroup relations and contrast social exclusion is highly needed.

Accumulated evidence has addressed the role played by intergroup contact in prejudice revision. In this respect, a wide range of studies has documented the effectiveness of direct, positive interactions with outgroup members in improving attitudes towards the outgroup (*Allport, 1954*; *Pettigrew & Tropp, 2006*). Not only direct contact but also indirect forms of contact, which do not require people to actually interact with each other, have shown to be effective in reducing prejudice towards different groups (*Lemmer & Wagner, 2015*;

Corresponding author
Soraya E. Shamloo,
sorayaelizabeth.shamloo@phd.units.it,
soraya.shamloo@gmail.com

*Brown & Paterson, 2016*). Likewise direct contact, indirect contact bases its effectiveness on the possibility of experiencing social interactions, may this be in the form of mental simulation, as in the case of imagined contact (e.g., *Crisp & Turner, 2009*), or in the form of media portrayals in which one is exposed to positive interactions between groups, as in the case of vicarious contact (e.g., *Mazziotta, Mummendey & Wright, 2011*).

In everyday life, people connect with others by relying on verbal but also on nonverbal communication, as in the case of physical contact. Touch in particular plays a pivotal role in human communication. Indeed, touch is at the basis of early human communicative interactions (*Field, 2001*; *Hertenstein et al., 2006a*; *Hertenstein et al., 2006b*) and develops into an elaborated symbolic system (*Jones & Yarbrough, 1985*). Touch can convey different emotions (*Hertenstein et al., 2006a*; *Hertenstein et al., 2006b*) and thus plays a major role in shaping interpersonal interactions (*Gallace & Spence, 2010* for a review). As far as physiological responses to touch are concerned, studies have shown that touch increases levels of oxytocin thus demonstrating its comforting and positive effects on well-being (e.g., *Holt-Lunstad, Birmingham & Light, 2008*). At the interpersonal level, the use of physical contact during interactions also promotes positive evaluations towards the toucher (*Erceau & Guéguen, 2007*), enhances cooperation within groups (*Kraus, Huang & Keltner, 2010*) and elicits pro-social behaviour (e.g., *Guéguen & Fischer-Lokou, 2003*). Within the literature on embodied cognition, empirical efforts were carried out to understand how aspects of social relationships are regulated by embodied cues, such as touch (*Gallese, 2001*; *Decety & Grèzes, 2006*; *Smith, 2008*; *Fantoni et al., 2016*). Based on this rationale, *Fiske (2004)* suggested that physical contact/touch among members is highly common within a specific form of relationship, namely the Communal Sharing Relationships. Communal Sharing Relationships are typical of close relationships in which group members share resources and focus on members' communality. These forms of relationships are said to be embodied by sharing among other things the use of physical proximity, touch and synchronized body movements (*Smith, 2008*) which may enhance cohesiveness among individuals (e.g., *Wiltermuth & Heath, 2009*).

Recently the impact of physical contact in intergroup relations has been partially addressed by research relying on the imagined intergroup contact paradigm (*Shamloo et al., in press*). In general, this paradigm requires that participants mentally simulate an intergroup encounter (*Crisp et al., 2010*). For instance, participants who were requested to imagine interacting with an outgroup member, reported lower levels of prejudice towards the outgroup as a whole than participants in a control condition (*Turner, Crisp & Lambert, 2007*). Research that relies on fMRi (i.e., functional magnetic resonance imaging) and PET (i.e., positron emission tomography) shows that mental imagery involves, at least in part, similar brain networks as those recruited in actual perception and emotion (*Kosslyn, Ganis & Thompson, 2001*). Based on this evidence, Turner and colleagues (2007) argue that the mental simulation of intergroup encounters allows the recruitment of mental structures that are present also during actual encounters with outgroup members (*Crisp et al., 2009*). In line with this insight, Hodson and colleagues (*2015*) required participants to imagine a cooperative interaction with a homeless person, in which physical contact was encouraged, or to simply imagine meeting a homeless person. Also, a neutral imagined scenario was

included in the experimental design as a control condition. Results indicated that, compared to the other conditions, only participants who imagined a cooperative interaction that involved a physical contact with a homeless person reported a weaker association between outgroup disgust and outgroup trust, which in turn mediated the relationship between disgust sensitivity and outgroup prejudice. Similarly, Choma and colleagues (*2014*) first assessed participants' prejudice, and then participants were asked to imagine physically interacting with an outgroup member. In this case the imagined physical encounter was based on team-building exercises (i.e., "thumb war" session), which required cooperation with the outgroup member. Before and after engaging in actual physical contact (i.e., wrist loops) with the outgroup member, participants' prejudice was assessed again. Results showed that following the imagined physical contact task, participants' levels of prejudice significantly decreased and remained stable across the following sessions.

The mentioned studies (*Choma, Charlesford & Hodson, 2014*; *Hodson, Dube & Choma, 2015*) suggest that physical contact may ameliorate outgroup attitudes. Nevertheless, these studies involved cooperation (e.g., trust-building or team-building exercises) between individuals which may have played a role in this respect (*Gaertner et al., 1990*). Said otherwise, as in this research the (imagined) physical encounter with an outgroup member was associated with a cooperative setting, and given that intergroup cooperation improves per se outgroup attitudes (*Gaertner et al., 1990*), the unique contribution of intergroup physical contact in ameliorating intergroup relations was only partially addressed by the above-mentioned research. To our knowledge, only a research carried out by Seger and colleagues (*2014*) has restricted its analyses to the impact of actual intergroup physical contact per se on outgroup attitudes. Specifically, Seger and colleagues (*2014*) tested the idea that a real touch, which is at the basis of Communal Sharing Relationships, would not only positively affect the evaluation of the individual toucher but also the group to which the toucher belongs.

In this respect, Seger and colleagues (*2014*) showed that a brief and casual touch (i.e., a single physical contact encounter) on the shoulder of European-American participants, performed by an African-American experimenter, reduced prejudice towards African-Americans, compared to when participants did not receive any touch. In other words, this research demonstrated that intergroup touch positively impacts on the evaluation of the outgroup toucher, and this positive experience generalizes to the toucher's group as a whole.

This research (*Seger et al., 2014*) underlines the potential role played by intergroup touch in reducing outgroup prejudice and opens up the question of why intergroup physical contact (i.e., physical contact between ingroup and outgroup members) contributes to the improvement of outgroup attitudes. Notwithstanding the importance of this research, the psychological mechanism that brings perceivers, who have experienced a physical encounter with an outgroup member, to improve their attitudes towards the outgroup as a whole has not been addressed yet. Said otherwise, the 'how' physical contact in intergroup relations shapes intergroup attitudes has not been investigated by previous research (*Choma, Charlesford & Hodson, 2014*; *Seger et al., 2014*; *Hodson, Dube & Choma, 2015*). The current research intends to fill this gap by analyzing the role of potential

mediating variables in the relation between intergroup physical contact and outgroup attitudes. In other words, we first intend to gather further evidence on the positive association between intergroup physical contact and outgroup attitudes thus strengthening the idea that enhanced levels of intergroup physical experience are associated with more positive attitudes towards the outgroup, and, more importantly, analyze the potential mediators that could account for the relation in question. To reach this aim, we analyze the relationship between individuals' amount of Experienced Intergroup Physical Contact (i.e., EIPC) with foreigners and attitudes towards this outgroup. Differently from Seger and colleagues' operationalization of intergroup physical contact, which involved a single casual touch, we assess participants' amount of EIPC, including a variety of types of physical contact involving different body parts and across a variety of contexts, thus relying on a comprehensive experience of physical contact with foreign individuals (i.e., outgroup). We hypothesize that the extent to which participants have experienced different intergroup physical contact encounters with outgroup members would work as the basis for the generalization of these positive experiences to the outgroup as a whole (Hypothesis 1). Second, we reasoned that intergroup physical encounters may facilitate intergroup contact which ultimately improves outgroup attitudes. As a case in point, recent research has shown that physical contact helps ameliorating close interpersonal relations (*Gulledge, Gulledge & Stahmannn, 2003*), backs pro-social behavior (*Guéguen & Fischer-Lokou, 2003*) and enhances cooperation (*Kraus, Huang & Keltner, 2010*). Also, physical contact is at the basis of cooperative relationships, in which individuals share a close interpersonal tie (*Fiske, 2004*) and create a sense of communality with others (*Seger et al., 2014*). If physical contact itself has the power of triggering such outcomes at the interpersonal level, it may be reasonable to think that it may exert similar effects at the intergroup level. Said otherwise, physical contact could enhance the perceived cooperation, pleasantness and the depth of intergroup encounters. These characteristics of an intergroup interaction are typically operationalized by the quality of intergroup contact. Hence, we would expect that higher levels of physical contact would predict better overall quality of intergroup contact. Not only physical contact might improve the quality of intergroup contact but it might also enhance the quantity of intergroup encounters. Indeed, it is plausible that the positive experience triggered by engaging in intergroup situations, in which also physical contact is involved, may enhance the opportunity of intergroup encounters by weakening the common avoidance-like reactions towards outgroup members (e.g., *Paladino & Castelli, 2008*; *Bianchi, Carnaghi & Shamloo, 2018*) and by encouraging more approach-like behaviors towards outgroup members, which ultimately impact on the frequency of intergroup encounters (*Kawakami et al., 2007*).

We thus hypothesize (Hypothesis 2) that higher levels of EIPC will be associated with a more positive appraisal of the intergroup contact (i.e., quality of intergroup contact) as well as enhanced opportunities of intergroup encounters (i.e., quantity of intergroup contact).

The distinct impact of quantity and quality of intergroup contact on different aspects of outgroup attitudes contributes to clarify the potential mediating role of these variables in the relation between the amount of EIPC and outgroup attitudes. Indeed, keeping these

two aspects of intergroup contact distinct would help to better understand which aspect of intergroup contact plays a major role in the relationship between intergroup physical contact and outgroup attitudes. This decision is supported by empirical evidence attesting to a different and distinct association between quality/quantity of intergroup contact, and outgroup attitudes and beliefs. Specifically, research which has focused on these two aspects of intergroup contact separately, has pointed out that quality, more so than quantity of intergroup contact, greatly predicts positive attitudes toward the outgroup (*Islam & Hewstone, 1993*; *Stephan, Diaz-Loving & Duran, 2000*; *Viki et al., 2006*). By contrast, quantity of intergroup encounters greatly impacts on perceived outgroup variability (*Islam & Hewstone, 1993*). Hence, quality, more so than quantity of intergroup contact seems to be associated with affective, evaluative-based reactions towards the outgroup (e.g., prejudice), while the frequency of intergroup contact is likely to be associated with the cognitive representation of the outgroup (e.g., outgroup homogeneity). For this reason, it might be plausible that the quality, rather than the quantity of intergroup contact would more likely predict participants' prejudice towards the outgroup. In the current study we assessed participants' evaluative-based reactions towards the outgroup by means of the General Evaluation Scale (*Wright et al., 1997*), which taps participants' evaluative responses to the target outgroup.

In sum, given that physical contact would positively predict the positive appraisal of intergroup encounters (quality of intergroup contact) as well as the frequency of intergroup encounters (quantity of intergroup contact), and due to the preferential association of quality over quantity of intergroup contact with prejudice, we hypothesized that quality of intergroup contact would be a solid candidate to mediate the association between the amount of EIPC and attitudes towards the outgroup. Therefore, we put forward that increased amounts of EIPC will be associated with more positive attitudes towards the outgroup because EIPC positively shapes the quality of the intergroup contact (Hypothesis 3).

## MATERIALS & METHODS

### Pilot study

Prior to the main research, a scale tapping participants' amount of experienced physical contact with known people (i.e., EPC-known person scale, see Appendix) was developed and taken into examination, thus allowing us to use the scale and adapt it to the purpose of the main study. A principle component analysis was performed in order to analyze its factor structure. In addition, we tested its reliability, and then its association with a proxy of physical contact, namely the Comfortable Interpersonal Distance scale (i.e., CID scale; (*Duke & Nowicki, 1972*) in order to test convergent and discriminant validity.

### Participants and procedure

Ninety-four participants ($n = 74$ female and $n = 19$ male participants, $n = 1$ did not report this information; age: $M = 20.53$, $SD = 3.99$) took part in the pilot study. Participants were told we were interested in collecting opinions regarding the social domain. Participants were provided a questionnaire and answered the following measures.
**Table 1** Factor loadings of the Principle Component Analysis on the EPC-known person items in the Pilot Study.

| Items | Component 1 |
|---|---|
| EPC-known person 1 | .75 |
| EPC-known person 2 | .80 |
| EPC-known person 3 | .83 |
| EPC-known person 4 | .68 |
| EPC-known person 5 | .45 |
| EPC-known person 6 | .72 |
| EPC-known person 7 | .67 |
| EPC-known person 8 | .79 |
| EPC-known person 9 | .51 |
| EPC-known person 10 | .73 |
| EPC-known person 11 | .62 |
| EPC-known person 12 | .65 |

**Notes.**
EPC-known person = experienced physical contact with a known person; EPC-known person from 1 to 12 refer to the twelve items used in the scale.

## Measures
### Amount of experienced physical contact with a known person (EPC-known person)

A 12-item scale tapping participants' amount of experienced physical contact with known people involving different body parts and regarding different situations was administered to participants. Participants rated the amount of experienced physical contact with a known person on a five-point scale, ranging from 1 (= *never*) to 5 (= *often*)[1]

### Comfort with interpersonal distance (CID)

Comfort with interpersonal distance was measured by using the Comfortable Interpersonal Distance scale (i.e., CID scale; *Duke & Nowicki, 1972*) with a known person (i.e., CID-known person) and a stranger (i.e., CID-stranger). Higher scores in this scale indicated lower comfort with interpersonal distance.

## RESULTS

We first performed a principal component analysis (Varimax Rotation) on participants' ratings of the items pertaining to the amount of experienced physical contact regarding known people. Results revealed a single factor structure that explained 47.7% of variance and factor loading ranged between .45 and .83 (see Table 1). Alpha could not be increased by eliminating any item. The reliability of the scale was high ($\alpha = .89$). Therefore, participants' ratings on the EPC-known person were averaged to reach a single composite measure of experienced physical contact ($M = 3.93$, $SD = .70$).

We then proceeded by testing the association of the EPC-known person scale with the CID-known person ($M = 1.11$, $SD = .64$) and CID-stranger scale ($M = 2.71$, $SD = 1.03$). As the CID scale in general is a measure of how comfortable people feel with interpersonal distance, it might represent a proxy of physical contact. Hence, participants' ratings on

[1] Participants also rated the EPC scale referring to unknown people (i.e., EPC-stranger), albeit this measure is beyond the scope of this pilot study.
the EPC-known person scale were regressed on their reactions to the CID-known person and CID-stranger scale. The CID-known person negatively predicted the EPC-known person ($\beta = -.27$, $t = 2.12$, $p = .04$), indicating that the more comfortable one felt with interpersonal distance with a known person, the higher the amount of physical contact experienced with known people. On the other hand, the CID-stranger positively predicted the EPC-known person ($\beta = .28$, $t = 2.15$, $p = .04$), showing that the less comfortable one felt with interpersonal distance with a stranger the higher the amount of physical contact experienced with known people.

To sum up, the EPC-known person scale has a good reliability and shows significant association with a proxy of physical contact and thus will be used in the main study.

## Study 1

Although past research (*Choma, Charlesford & Hodson, 2014*; *Seger et al., 2014*; *Hodson, Dube & Choma, 2015*) has focused on an experimental approach to study the effects of intergroup physical contact on attitudes towards the outgroup, in the current study we tackle this issue by using a correlational approach for two distinct, albeit related reasons. Firstly, this method has been largely used in research addressing the relationship between intergroup contact and outgroup attitudes, and constitutes a reliable approach to the study of social psychological processes involved in the intergroup contact (e.g., *Voci & Hewstone, 2003*; *Turner, Hewstone & Voci, 2007*; *Christ et al., 2010*). Secondly, the current work aims to study the frequency of different types of physical contact involving different body parts and across a variety of contexts. In other words, we intend to capture a broad and comprehensive experience of physical contact with foreign individuals (i.e., outgroup), rather than a single intergroup touch (*Seger et al., 2014*), across different everyday life contexts, rather than in a specific context (*Choma, Charlesford & Hodson, 2014*; *Hodson, Dube & Choma, 2015*). For this reason, a survey would match the current aim as it does not constrain the analyses to a limited number and types of physical experiences as well as contexts in which the intergroup physical experiences occurred. Although relying on a cross-sectional design allows us to explore how intergroup physical contact relates to outgroup attitudes in a natural setting, thus enhancing the ecological validity of our findings, it prevents us from strong claims about the causal relationship among variables.

### Participants and procedure

We decided to rely on a sample of 100 participants. This decision was backed by a sensitivity analyses (G Power 3.1; *Faul et al., 2007*), $\alpha$ err. prob. $= .05$, Power ($1 - \beta$ err. prob.) $= .8$, $N = 100$, which indicated a Minimal Detectable Effect (MDE) size f $= .11$. Hence, the smallest *real* effect size which we would be able to detect (at 80% power) with this sample size falls within the small-effect size area (*Cohen, 1988*). We decided to collect more than 100 participants to be sure to reach the estimated sample size given the probability of coming across missing data. One hundred eleven students from a university in northern Italy participated in this study. Two participants did not report their gender and could not be entered in the *lme* model which treated this variable as a covariate. The remaining participants did not fully rate the variables included in the mediation models ($n = 1$ on

the EIPC scale; $n = 2$ on quality of intergroup contact; $n = 1$ on the prejudice scale; $n = 1$ on both the prejudice scale and quality of intergroup contact). Given that we relied on aggregated measures of participants' ratings as indexing the variable under examination, the exclusion of the participants who did not rate all the items of a given scale is needed as calculating the synthetic value of the scale on a different and limited number of rated items would undermine the reliability of the entire measures (Supplemental Information 1). The final sample comprised $n = 58$ female participants and $n = 46$ male participants (age: $M = 22.12$, $SD = 2.99$). Among these participants, $n = 98$ were Italians, $n = 5$ were not Italians and $n = 1$ indicated two nationalities. The current sample size approximated the N rule.

Prior to filling in the questionnaire, the researcher provided participants with the written consent and assured they had understood it.

We decided to rely on the outgroup target 'foreigners' for different reasons. First, Asbrock and colleagues (2014) found that when participants are asked to think about foreigners, they tend to indicate and think about the most salient ethnic minorities in a given country (p. 6, Asbrock et al., 2014), thus suggesting a large overlap between the term 'foreigners' and ethnic minorities. In line with the above-mentioned claims, several studies rooted in the intergroup relation tradition have often measured prejudice towards foreigners in general (e.g., Liebkind & McAlister, 1999; Raijman, Semyonov & Schmidt, 2003; Christ et al., 2010). Second, the term 'foreigners', at least in the Italian context, is typically used as synonym of nonItalian, ethnic minorities and recently employed by the National Institute of Statistics to assess Italian respondents' attitudes towards ethnic minorities (Istituto Nazionale di Statistica, 2018).

Quantity and quality of intergroup contact, the amount of EIPC with foreign people and outgroup attitudes were self-assessed. Half of the participants rated the measures in the above-mentioned order, whereas the other half rated outgroup attitudes first, then quantity and quality of intergroup contact and then the amount of EIPC with foreign people (i.e., order of presentation).

## Measures
### Quantity and quality of contact with foreign people
Two items (adapted from Voci & Hewstone, 2003) tapped the quantity of intergroup contact ($\alpha = .71$), namely: "How many foreign people do you know?" (None-More than 10), "How frequently do you have contact with foreign people?" (Never-Very frequently). All answers were given on a five-point scale. As for the quality of intergroup contact ($\alpha = .75$; adapted from (Voci & Hewstone, 2003), participants were asked: "When you meet foreign people, in general you find the contact…" and presented with three adjectives: pleasant (piacevole in Italian), cooperative (cooperativo in Italian), superficial (superficial in Italian). Answers ranged from 1 (=Not at all) to 5 (=Totally). As an aggregated measure of the quantity of intergroup contact and quality of intergroup contact, we relied on the median value of the corresponding items.

### Amount of experienced physical contact with foreign people

Given the single factor structure yielded by the EPC- known person scale used in the pilot study, its good reliability and association with a proxy of physical contact (i.e., the comfortable interpersonal distance), the EPC-known person scale was adapted to assess the amount of EPC with foreign individuals in particular ($\alpha = .91$). Participants' ratings were averaged to form a synthetic score of EPC with foreign individuals.

### Attitudes toward foreign people

Participants were asked to "describe how you feel when thinking about foreign people in general" by using six bipolar adjectives (e.g., warm/cold) on a seven-point scale ($\alpha = .88$; (*Wright et al., 1997*). Participants' ratings were averaged to form a general score of intergroup prejudice. Higher values indicated a more positive attitude towards the outgroup.

At last, participants reported their gender, age, nationality and native language. Participants were then thanked and fully debriefed. This study was carried out in accordance with the Ethical Committee of the University of Trieste (approval number 84) and in accordance with the declaration of Helsinki.

## RESULTS

To verify whether higher amounts of EIPC predicted more positive intergroup attitudes we performed a causal mediation analysis. The EIPC to intergroup attitudes relationship exhibited both a direct and an indirect pathway through quality and/or quantity of contact with foreign people. We extracted these pathways together with the indices of their statistical reliability, using the mediation R software and performing a causal mediation based on linear mixed effect as mediator model types. In particular path coefficients (i.e., $\beta \pm 1$ standard error of the mean) were estimated using linear mixed effect models fitted by minimizing the restricted maximum likelihood criterion (*Laird & Ware, 1982*). The advantages of this type of models over the more traditional one based on mixed-model ANOVAs is discussed by Kliegl and colleagues (*2010*). In particular, *lme* was shown to be more robust to unbalanced dataset and to suffer less severe loss of statistical power compared to mixed-model ANOVAs. We followed *Bates (2010)* and used this statistical procedure to obtain two-tailed *p*-values by means of likelihood ratio test based on $\chi^2$ statistics when contrasting *lme* with different complexities. Furthermore, we used type 3-like two tailed *p*-values for significance estimates of *lme*'s fixed effects and parameters adjusting for the *F*-tests the denominator degrees-of freedom with the Satterthwaite approximation based on SAS proc mixed theory. Finally, as indices of effect size, of the predictive power and of the goodness of fit for the relevant paths estimated through *lme* models, we selected the Pearson-$r^2$ and the concordance correlation coefficient, the $r_c$. According to *Vonesh, Chinchilli & Pu (1996)*; but see also (*Rigutti, Fantoni & Gerbino, 2015*) this latter index provides an optimal measure of the degree of agreement between the observed values and the lme predicted values, in the $-1$ to 1 range. As an additional measure of significant effect size associated to *lme* estimated coefficient, we provided Cohen's *d*. To implement our mediation analysis, we used a default simulation type quasi-Bayesian

Monte Carlo method based on normal approximation (*Imai, Keele & Tingley, 2010*). In addition, a bootstrapping method with a number of re-samples large enough to guarantee reliable results (i.e., 2000) was used to compute confidence intervals of the proportion of effect mediated by quality and/or quantity of contact as inferred from the average causal mediation (ACME), average direct (ADE) and average total effects. The main mediation model we tested resulted from two preliminary analyses, contrasting competing models with increasing complexities (i.e., degrees of freedom), but with the same random component. In particular, in both analyses participant gender and nationality were treated as both fixed and random intercepts throughout our analysis for two major reasons: (1) we did not have any specific predictions on the way gender and nationality might affect the EIPC to intergroup attitudes relationship; (2) we aimed to maximize the robustness of the mediation analysis over individual variability.

In the first analysis a comparison of *lme* models with nested fixed effects showed that the outcome variable was not affected by our balancing variable (i.e., order of presentation, $\chi^2_1 = 0.02$, $p = .90$). We thus excluded such a factor from the remaining analyses. The second preliminary *lme* analysis was aimed at identifying the statistical structure of the *lme* model that best represented the EIPC to intergroup attitudes relationship. The best representative structure indeed might be characterized by either a multiplicative model including all interaction terms or a simpler additive model including only the main effects. We selected the best model amongst our two competing models thus entering the amount of EIPC as the predictor variable, quality and quantity of intergroup contact as two independent mediators, and outgroup attitudes as the outcome variable, with participant gender and nationality treated as both fixed and random effects. Importantly, contrasting the two models, no interaction term turned out to be significant ($\chi^2_{16} = 22.43$, $p = .13$) and we thus proceeded by performing the causal mediation analysis treating our predictors as independent factors with gender and nationality not significantly affecting the outcome variable ($F_{1,98} = 0.30$, $p = .58$ for nationality; $F_{1,98} = 1.64$, $p = .20$ for gender). This model accounted for a significant portion of the outcome variance ($r^2 = 0.34$, $r_c = 0.51$, 95% CI [0.38 - 0.62], $F_{5,98} = 10.28$, $p < .001$).

The model, shown in Fig. 1, revealed a significant *Total Effect*, with higher amounts of EIPC associated with more positive outgroup attitudes (consistent with H1, $\beta = 0.41 \pm 0.12$, $df = 100$, $t = 3.49$, $p < .001$, $d = 0.70$, $r^2 = 0.18$, $r_c = 0.304$, 95% CI [0.179–0.419]). Importantly, the amount of EIPC contributed to the variance of outgroup attitudes, and also contributed to the variance of both quality ($r^2 = 0.25$, $r_c = 0.40$, 95% CI [0.27–0.51], $F_{1,100} = 20.48$, $p < .001$) and quantity ($r^2 = 0.38$, $r_c = 0.55$, 95% CI [0.43–0.65], $F_{1,100} = 54.10$, $p < .001$) of intergroup contact (consistent with H2). Also, quality ($r^2 = 0.33$, $r_c = 0.50$, 95% CI [0.37–0.61], $F_{1,100} = 37.28$, $p < .001$) and quantity ($r^2 = 0.14$, $r_c = 0.24$, 95% CI [0.12–0.35], $F_{1,100} = 6.75$, $p < .011$) affected the outcome variable. Furthermore, outgroup attitudes improved as quality ($\beta = 0.59 \pm 0.10$, $df = 100$, $t = 6.10$, $p < .001$, $d = 1.22$) and/or quantity ($\beta = 0.27 \pm 0.10$, $df = 100$, $t = 2.53$, $p < .01$, $d = 0.51$) of intergroup contact increased, which in turn enhanced with the increase of the amount of EIPC (quality: $\beta = 0.45 \pm 0.10$, $df = 100$, $t = 4.53$, $p < .001$, $d = 0.91$; quantity: $\beta = 0.69 \pm 0.09$, $df = 100$, $t = 7.35$, $p < .001$, $d = 1.47$). Such a pattern of mutual

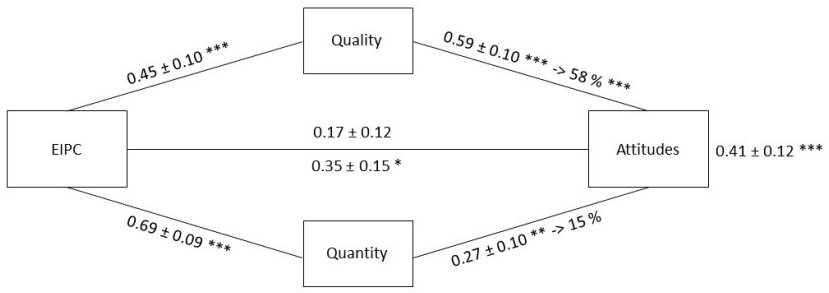

**Figure 1 Causal *lme* mediation analysis on attitudes towards foreigners with the amount of experienced intergroup physical contact (EIPC) as predictor variable, and quality and quantity of intergroup contact as mediators (assessed independently).** Coefficients marked with one, two or three asterisks are significant at $p < .05$, $p < .01$ and $p < .001$ level, respectively. The effects of the predictor variable on the mediators (i.e., quality and quantity of intergroup contact) are shown on the arrow lines connecting EIPC to the mediators. The contribution of the mediators to outgroup attitudes is depicted on the line connecting the mediators to outgroup attitudes. The *lme* estimates of the *Total Effect* of the predictor variable on attitudes is included in the rightmost part of the model next to the Attitudes box. The *Direct effects* (with quality and quantity as mediators) are depicted in the middle part of the model. The proportion of effect mediated by quality and quantity of intergroup contact is depicted above the line connecting the mediators to attitudes, following the arrows.

relationships provided a strong ground for a potential mediation of the total effect by quality and/or and quantity of intergroup contact. The mediating role of quality of intergroup contact was attested by the fact that when it was added as a covariate to the effect of the amount of EIPC on outgroup attitudes, the direct association (i.e., Direct Effect) between the amount of EIPC and outgroup attitudes significantly decreased, reaching a coefficient statistically equal to zero ($\beta = 0.17 \pm 0.12$, $df = 99$, $t = 1.48$, $p = .14$). Also, no significant loss in the fit was found when contrasting an *lme* model with quality as the only fixed effect vs. an *lme* model including all fixed effects (with $r_c$ slightly decreasing from 0.51, 95% CI [0.38–0.62] to 0.50, 95% CI [0.37–0.60], $\chi^2_1 = 2.29$, $p = .13$). By contrast, no such mediating effect was found for quantity of intergroup contact. No reliable modification of the direct association between the amount of EIPC and outgroup attitudes was found when quantity was included as a mediator ($\beta = 0.35 \pm 0.15$, $df = 99$, $t = 2.37$, $p = .02$, $d = 0.48$), with a significant loss in the fit when contrasting an *lme* model with quantity as the only fixed effect vs. an *lme* model including all fixed effects (with $r_c$ increasing from 0.24, 95% CI [0.12–0.35] to 0.31, 95% CI [0.19–0.43], $\chi^2_1 = 5.74$, $p = .02$). The *Total Effect* was accounted for by quality of intergroup contact (consistent with H3), with a significant proportion of mediation (0.58, 95% CI [0.27–1.21], $p < .001$) supported by the highly reliable Average Causal Mediation Effect (ACME = 0.24, 95% CI [0.11–0.40], $p < .001$). No such mediation was found for quantity of intergroup contact (proportion mediated = 0.15, 95% CI [−0.30–0.70], $p = .49$; ACME = 0.06, 95% CI [−0.12–0.24], $p = .49$).

To better ascertain the directionality of the relationship suggested by the previous analysis, we proceeded by comparing the goodness-of-fit of the above specified mediation model (i.e., Model 1) with that of an alternative mediation model, by inverting the directionality of the relationship between the mediators and the predictor (i.e., Model 2).

In this alternative model only the *Total Effect* specified by the relation between quality of contact and outgroup attitudes was significant ($\beta = 0.56 \pm 0.10$, $df = 104$, $t = 5.53$, $p < .001$, $d = 1.08$; $r^2 = 0r^2$, $r_c = 0.50$, 95% CI [0.37–0.61], $F_{1,104} = 30.6$, $p < .001$). However, when the amount of EIPC was entered as a mediator, this relationship still remained significant ($\beta = 0.53 \pm 0.10$, $df = 98$, $t = 4.89$, $p < .001$, $d = 0.99$; $r^2 = 0.34$, $r_c = 0.51$, 95% CI [0.38–0.62], $F_{1,98} = 23.91$, $p < .001$), thus proving that the amount of EIPC did not mediate the relation between quality of contact and outgroup attitudes.

We further compared the two models. To attain this aim, we contrasted the two models by using the Structural Equation Analyses (SEA), Lavaan R software (*Rosseel, 2012*). This analyses allowed us to quantify the differential fit of the two models by means of distinct and different indices, such as $\chi^2/df$ ratio, SRMR, CFI and AIC. It is worth noting that, and according to *Hu & Bentler (1999)*, acceptable fit is revealed by a $\chi^2/df$ ratio of less than 3, SRMR less than .08, and a CFI greater than or equal to .95; comparative measure of fit is also given by lower levels of AIC in a model over the alternative model. Based on the analyses of these parameters, Model 1 showed a good fit of the data ($\chi^2 = 2.3$, $df = 1$, $p = .61$; SRMR = .07; CFI = .97; AIC = 954.8), and a higher goodness-of-fit than Model 2 ($\chi^2 = 15.52$, $df = 1$, $p = .05$; SRMR = .15, CFI = .85; AIC = 975.7).

## DISCUSSION

This study aims at understanding the role played by intergroup physical contact in shaping attitudes towards foreigners, and at testing the mediating role of intergroup contact (i.e., quantity and quality) in this respect.

Results indicate that higher amounts of EIPC are associated with more positive outgroup attitudes. This result confirms Seger and colleagues' (*2014*) findings, and suggests that the effect of physical contact goes beyond a brief and casual touch. Indeed, the EPC scale allowed us to assess individual differences in the amount of EIPC by capturing broader and detailed aspects of body-based encounters. Moreover, the amount of EIPC likely facilitates intergroup encounters, as testified by the fact that as the amount of EIPC increased, the frequency of intergroup interactions also increased. In addition, the amount of EIPC was also associated with the perceived quality of the intergroup contact. Indeed, higher amounts of EIPC were linked to more pleasant, less superficial and more cooperative intergroup interactions. Importantly, and in line with our hypothesis, only quality, and not quantity of intergroup contact, mediated the relation between the amount of EIPC and outgroup attitudes.

However, one may claim that quantity, and quality of intergroup contact in particular, would be linked to the amount of EIPC (and not vice versa), which in turn would mediate the relation between intergroup contact and outgroup attitudes. Indeed, intergroup contact in general, and quality of intergroup contact in particular, has been found to be predictive of social distance (*Binder et al., 2009*), which is also related to physical distance (*Brewer, 1968*). We directly tested this alternative model (i.e., Model 2), which showed a lower goodness-of-fit than the hypothesized one (i.e., Model 1). Moreover, the comparison between the two models showed that quality of intergroup contact mediated the relation

between the amount of EIPC and outgroup attitudes, while the amount of EIPC did not mediate the relation between quality of intergroup contact and outgroup attitudes. Hence, we may suggest that the amount of EIPC likely plays a role in shaping the appraisal of intergroup encounters, since the reverse relation was not supported by our data.

## CONCLUSIONS

These results suggest that intergroup physical encounters may facilitate intergroup contact, and therefore should be taken into account when discussing about strategies aimed at ameliorating outgroup attitudes. Also, these results add further support to the existing relation between intergroup physical contact and outgroup attitudes (i.e., *Choma, Charlesford & Hodson, 2014*; *Seger et al., 2014*; *Hodson, Dube & Choma, 2015*; *Shamloo et al., in press*), and for the first time shed light on the mediating variables involved in this relation. We do believe these results may raise awareness on how physical contact may represent a way to facilitate more pleasant relationships with the individuals we communicate with, by enhancing the quality of these interactions and then improving attitudes towards the group to which these individuals belong. This pattern of results opens up to future studies which might experimentally vary the quantity of intergroup physical contact (i.e., high frequency of physical contact vs. low frequency of physical contact), assess a general appraisal of future intergroup contact, and test whether the variation in the frequency of intergroup physical contact shapes outgroup attitudes because it improves the anticipated quality of the intergroup contact.

Nevertheless, some limits should be acknowledged. First, this is a correlational study and we reckon this study to be exploratory. Although these results hint to the fact that intergroup physical contact improves the quality of intergroup encounters, future studies should test this hypothesis also by using a longitudinal approach and/or an experimental design, thus ascertaining the causal direction of the variables in question.

Second, when considering participants' physical contact experience, their disposition towards engaging in and receiving physical contact should be assessed (*Webb & Peck, 2015*). Also, given the cultural differences in terms of preferred interpersonal distances (*Sorokowska et al., 2017*), willingness to engage in physical contact, and in the meaning associated with this type of encounters (*Remland, Jones & Brinkman, 1995*), future studies should test the proposed model in different cultural settings, thus strengthening its external validity. Despite the significance of our findings more works are needed, especially experimental research is requested to test the validity of the presented model.

[2]In the EPC scale the "X" was replaced with different target groups for the pilot (X = known person) and for the main study (X = foreign person).

## APPENDIX

How many times have you **held the hand** of a X[2] ?
How many times have you **walked arm in arm** with a X?
How many times have you **caressed** a X?
How many times have you **got a massage** from a X?
How many times have you **got a hair wash** by a X?
How many times have you **put your hand on the shoulder** of a X?

How many times have you **sat on the knees** of a **X**?
How many times have you **hugged** a **X**?
How many times have you **shaken hands** with a **X**?
How many times have you **kissed** a **X**?
How many times have you **stood shoulder to shoulder** with a **X**?
How many times have you **high-fived** a **X**?

### Funding

This research was supported by FRA Finanziamento per Ricerca di Ateneo (FRA 2016 University of Trieste). There was no additional external funding received for this study. The funders had no role in study design, data collection and analysis, decision to publish, or preparation of the manuscript.

### Grant Disclosures

The following grant information was disclosed by the authors:
FRA Finanziamento per Ricerca di Ateneo (FRA 2016 University of Trieste).

### Competing Interests

The authors declare there are no competing interests.

### Author Contributions

- Soraya E. Shamloo conceived and designed the experiments, performed the experiments, analyzed the data, prepared figures and/or tables, authored or reviewed drafts of the paper, approved the final draft.
- Andrea Carnaghi conceived and designed the experiments, analyzed the data, authored or reviewed drafts of the paper, approved the final draft.
- Carlo Fantoni analyzed the data, authored or reviewed drafts of the paper, approved the final draft.

### Human Ethics

The following information was supplied relating to ethical approvals (i.e., approving body and any reference numbers):

The study was approved by the Ethical Committee of the University of Trieste (approval number 84).

### Data Availability

The raw data are provided in the Supplemental Files.

### Supplemental Information

Supplemental information for this article can be found online at http://dx.doi.org/10.7717/peerj.5680#supplemental-information.

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
