# Peer review of "Investigating the relationship between intergroup physical contact and attitudes towards foreigners: the mediating role of quality of intergroup contact"

_PeerJ, doi:10.7717/peerj.5680_

## Round 0.1 · original submission · Major Revisions

I have now received two reviews on your manuscript. I thank the reviewers for their work. As you will see, both reviewers think that your manuscript addresses an important topic, but both of them express numerous concerns. While Reviewer 2 proposes “major revision”, Reviewer 1 proposes to reject the paper. After my carefully reading of your paper, I decided to leave the door open for a resubmission. However, the revised paper will have to convince both the reviewers and myself. In the rest of my action letter let me focus on some issues that I consider crucial for a successful revision. For the details please refer to the reviews, that are thoughtful and quite clear.

Better grounding in current literature. I think you should make an effort to better ground your work in the framework of recent literature, clarifying the eventual gaps you are going to fill/address with your work (see comments of both reviewers).

The terminological choice should be further justified (see comments of Reviewer 2)

The reasons underlying your choice of the methodology and of the sample should be further detailed, in particular you should explain why you decided to perform a survey (see comment of Reviewer 1) and why you focused on migrants without providing specifications (see comment of Reviewer 1)

Further empirical evidence: I agree with Reviewer 1 that an experimental confirmatory study that replicates the results of the reported study would make the paper much stronger.

Thank you for sending your interesting work to PeerJ.

Reviewer 1 ·

Basic reporting

This paper examined with a single study whether the Experienced Intergroup Physical Contact (EIPC) increase positive attitudes towards an outgroup (i.e. foreigners) by mean of the quality of the intergroup physical contact.

The paper touches a topic highly relevant and timely: prejudice towards foreigners and ways to reduce it. It is well written and clear. The structure conforms to the standard sections in a scientific paper. It includes a table and a figure and the authors share raw data.
However one of the main weaknesses of it is that it did not review highly relevant prior research about the main topic of the manuscript, as for example the work done by Choma, Charlesford & Hodson (2014) or Hodson, Dube & Choma (2015). Furthermore, in the introduction nothing is said about the extensive literature on embodied cognition that is also relevant when analysing the relation between physical contact and prejudice. Therefore, it is not clear if the current study adds enough to the field. It is seems that the most important point that the paper adds to the prior research has to do with one of the mechanisms that could explain why physical contact can improve intergroup attitudes: the quality of this physical contact. However, the authors predicted upfront this mediation but there is hardly any justification of this prediction, therefore it is seemed more a postdoc one.

Another important pitfall of the paper has to do with the lack of specification about who were the participants thinking about when they were asked about foreigners. From my point of view it is not the same to think about immigrants who move to find better life conditions in a new country and are usually targets of prejudice than about high class tourists who travel for holidays but are usually by natives. From my point of view it would be much more interesting if the research would have focused just in negatively evaluated outgroups.

Minor point:
The format of the figure seems a bit odd. It is too big and uses colours to show the coefficients of each path, however usually figures for mediation analyses used black and white graphs.

Experimental design

The paper fits well within the aims and scope of the journal. The research question is properly defined, but as I said previously, although authors posited a knowledge gap that the paper tries to fill, this gap it is not clear to me if you take into account the previous work (including the one not reviewed by the authors).
The research fits with ethical standards in the field.
The method described is enough to allow replication, but the measures used have a limited scope as the scale used to measure experienced physical contact is built for the present research and not adequately validated.

Validity of the findings

The results seems to be statistically sound, however the manuscript presents just one main study and its hypotheses were not pre-registered and it did not show a neither a prior power analysis to determine the sample size nor a post-doc one in order to determine the power of the study given the sample size used.
It just presented a correlational study (plus a pilot one), and therefore the conclusions are very limited, although the author acknowledged it. The pilot study seems to have the goal of building and validating the EIPC, however the authors did not perform the standard process of scale building and validation.

Additional comments

In my opinion the paper would add significantly more to the field if it includes an experimental confirmatory study that replicates the results of the mediation showed in the main study of the paper.

·

Basic reporting

Reporting is ambiguous and its difficult to understand.
1. The manuscript is replete with stylistic errors and it would benefit from some proofreading.
2. Confusion of touch vs contact
3. Race or ethnicity? “an African-American experimenter, reduced prejudice towards African-Americans, compared to (line 67) when participants did not receive any touch. In other words, this research demonstrated that 68 interethnic

Experimental design

The paper employs survey design but it does not discuss and convince the audience why is this the best method to investigate the topic. I was under the impression that this was manuscript based on experimental design as they refer to “experimenter”. On a separate note I had a real struggle to understand the terminology the authors use to report their results. for instance (an in no particular order):
1. What does it mean and what is “A causal mediation analysis using linear mixed effect (i.e., lme, fitted by minimizing the (line 200) restricted maximum likelihood criterion, Laird & Ware, 1982; see Rigutti, Fantoni, & Gerbino,201 2015, for a similar analysis) as mediator model types was performed to verify whether higher”
2. Did the authors have an predictions regarding these? “used as two independent mediators, and outgroup attitudes as the (line 206) outcome variable, with participant gender and nationality treated as both fixed and random (line 207) effects in order to maximize the robustness of the mediation analysis over individual variability”
3. Why 2000 resamples?
4. What are the effect sizes for individual paths?
5. What platform did the authors use to analyse their data?
6. The model fit comparison appears to be a qualitative one as I don t see the results of a chi square comparison test?

Validity of the findings

The question itself is interesting but is there gap in the literature that the manuscript addresses? How does the manuscript relate to the previous research and extend it? The introduction summarizes the literature on various themes of intergroup contact but I do not think it goes beyond replication of previous research, in fact only one paper, but it fails to identify the gap in this literature and to introduce the research question. What is it the authors want to show or claim? We should create conditions to facilitate people touching each other?
Why did the authors exclude the following participants?
Participants who did not answer to more than one item on the prejudice scale (n = 1) or who did (line 160) not report relevant information such as gender (n = 2) were excluded from the sample.

Additional comments

I read the manuscript titled “Investigating the relationship between intergroup physical contact and attitudes towards foreigners:the mediating role of quality of intergroup contact” ". The study investigates the impact of physical contact on outgroup attitudes via quality and quantity of contact. Given the popularity of intergroup contact theory this is an interesting area of research. As such, the manuscript has the potential to inform our understanding of the link between ideology and praxis with significant implications for education. Sadly, I do not think that this potential has been realized and I have certain concerns that prevent me from me from supporting the publication of the manuscript. I outline these below.

Theoretical Novelty/Clarity: The question itself is interesting but is there gap in the literature that the manuscript addresses? How does the manuscript relate to the previous research and extend it? The introduction summarizes the literature on various themes of intergroup contact but I do not think it goes beyond replication of previous research, in fact only one paper, but it fails to identify the gap in this literature and to introduce the research question. What is it the authors want to show or claim? We should create conditions to facilitate people touching each other?
Empirical Contributions & Methodology: The paper employs survey design but it does not discuss and convince the audience why is this the best method to investigate the topic. I was under the impression that this was manuscript based on experimental design as they refer to “experimenter”. On a separate note I had a real struggle to understand the terminology the authors use to report their results. for instance (an in no particular order):
1. What does it mean and what is “A causal mediation analysis using linear mixed effect (i.e., lme, fitted by minimizing the (line 200) restricted maximum likelihood criterion, Laird & Ware, 1982; see Rigutti, Fantoni, & Gerbino,201 2015, for a similar analysis) as mediator model types was performed to verify whether higher”
2. Did the authors have an predictions regarding these? “used as two independent mediators, and outgroup attitudes as the (line 206) outcome variable, with participant gender and nationality treated as both fixed and random (line 207) effects in order to maximize the robustness of the mediation analysis over individual variability”
3. Why 2000 resamples?
4. What are the effect sizes for individual paths?
5. What platform did the authors use to analyse their data?
6. The model fit comparison appears to be a qualitative one as I don t see the results of a chi square comparison test?
Other Issues:
1. The manuscript is replete with stylistic errors and it would benefit from some proofreading.
2. Confusion of touch vs contact
3. Race or ethnicity? “an African-American experimenter, reduced prejudice towards African-Americans, compared to (line 67) when participants did not receive any touch. In other words, this research demonstrated that 68 interethnic
4. I am utterly confused by the claim that the “Quantity and quality of intergroup (line 99) contact have often been treated together, we reckon that dissociating these two aspects of” especially when this claim is followed by a review of research on the differential effects of quality vs quantity of contact


Why did the authors exclude the following participants?
Participants who did not answer to more than one item on the prejudice scale (n = 1) or who did (line 160) not report relevant information such as gender (n = 2) were excluded from the

---

## Round 0.2 · Minor Revisions

I apologize for my delay. I ask you to address the last issues raised by the reviewer; next time I will handle the paper myself without further re-review.

Reviewer 1 ·

Basic reporting

I appreciate the authors’ efforts to respond to each of our comments, most of which address the relevant concerns.

On the theoretical side, I think the manuscript in its current form does much better at discussing the literature that was not included in the first version of the manuscript and highlights the gap that the study intended to fill within it, specially regarding the contribution of intergroup physical contact in the field of intergroup attitudes and the mechanisms which can explain it. Furthermore, they now make much more explicit why they predict that quality, and not just quantity of intergroup contact, may mediate the relation between intergroup physical contact and intergroup attitudes.

Finally, still some typos and grammatical errors remain in some of the revised parts of the manuscript.

Experimental design

Methodologically, I am satisfied with the sensitivy analyses run in order to justify the sample size. However, from my point of view is not convincing the reasons why the authors decided to exclude some participants from the analyses (they said because they do not answer with “relevant information”). Therefore I strongly recommend to show the results without any exclusions. If the results remain the same, they would not have any problem, if this were not the case, they would have to try to explain the differences. In my view the authors solved the rest of the most important methodological issues I raised in my previous review.

Validity of the findings

That said, and although I am aware of the difficulty to gather more data from a confirmatory or experimental study, they should acknowledge this point in the limitations of the manuscript, specially the exploratory nature of their results.

Additional comments

Overall, I think this is an interesting set of findings indicating that physical intergroup contact could improve intergroup attitudes through the improvement of the quality of intergroup contact. However, in the manuscript it should be clear that this is just a first step and some more should be done in order to confirm this result and deep on causal relations between the relevant variables.

---

## Round 0.3 · accepted · Accept

I am happy to inform you that your manuscript has been accepted for publication on PeerJ.

#